# Null Mismatch Repair Proteins Expression Reveals the Temporal Molecular Events in Lynch Syndrome-Related Cancers

**DOI:** 10.3390/diagnostics14090888

**Published:** 2024-04-24

**Authors:** Gianmaria Miolo, Wally Marus, Angela Buonadonna, Lucia Da Ros, Lara Della Puppa, Giuseppe Corona

**Affiliations:** 1Medical Oncology and Cancer Prevention Unit, Centro di Riferimento Oncologico di Aviano (CRO), IRCCS, 33081 Aviano, Italy; abuonadonna@cro.it (A.B.); luciadaros@cro.it (L.D.R.); 2Pathology Unit, Department of Medicine Laboratory Section, Pordenone Hospital, 33170 Pordenone, Italy; wally.marus@asfo.sanita.fvg.it; 3Oncogenetics and Functional Oncogenomics Unit, Centro di Riferimento Oncologico di Aviano (CRO), IRCCS, 33081 Aviano, Italy; ldellapuppa@cro.it; 4Immunopathology and Cancer Biomarkers Unit, Centro di Riferimento Oncologico di Aviano (CRO), IRCCS, 33081 Aviano, Italy; giuseppe.corona@cro.it

**Keywords:** MMR genes, *MSH2* gene, immunohistochemical analysis, null phenotype

## Abstract

The immunohistochemical assessment of mismatch repair (MMR) proteins represents a pivotal screening tool for identifying Lynch syndrome (LS)-related cancers, as the loss of their expression often indicates MMR dysfunction associated with genetic or epigenetic alterations. Frequently, LS-related colorectal cancers present germline pathogenic variants in the *MLH1* or *MSH2* genes, which result in the simultaneous immunohistochemical loss of MLH1 and PMS2 or MSH2 and MSH6 proteins expression, respectively. Less commonly observed is the single involvement of the MSH6 or PMS2 proteins expression, indicative of the presence of germline pathogenic variants in the corresponding genes. Extremely rarely reported are the null immunohistochemistry phenotypes represented by the complete loss of expression of all MMR proteins. The molecular mechanisms contributing to the raising of this latter uncommon immunohistochemical phenotype are derived from the combination of pathogenic germline variants in MMR genes with the somatic hypermethylation of the *MLH1* gene promoter. This study focuses on elucidating the molecular cascade leading to the development of the null immunohistochemical phenotype, providing valuable insights into understanding the sequential molecular events driving the LS-associated tumorigenesis, which may have pivotal implications in the clinical management of patients with LS-related cancers.

## 1. Introduction

Mismatch repair (MMR) proteins play a crucial role in maintaining genomic stability during DNA replication and recombination [1,2]. The MMR system consists of several proteins, including MLH1 (MutL homolog 1), MSH2 (MutS homolog 2), MSH6 (MutS homolog 6), and PMS2 (postmeiotic segregation increased 2), which, in an orchestrated manner, recognize and repair DNA mismatches arising during DNA replication [3]. Thus, dysfunction in MMR proteins can lead to genomic instability predisposing individuals to the development of various cancers. In clinical practice, tumor characterization of MMR proteins expression is typically performed through immunohistochemistry (IHC) analysis, which provides valuable information about their expression and subcellular localization. While normal tumor MMR proteins expression is characterized by strong nuclear staining, their absence or reduction in the expression suggests potential defects in the MMR system [3,4]. To date, the IHC characterization of MMR proteins expression represents a pivotal screening tool in the identification of patients with suspected Lynch syndrome (LS), a cancer predisposition syndrome inherited in an autosomal dominant manner and due to germline pathogenic variants in MMR genes [3,5,6,7]. LS-related cancers often display a characteristic loss of expression in MMR proteins that predominantly involves the MLH1 and PMS2. This loss is commonly attributed to the high prevalence of germline variants within the *MLH1* gene that disrupt the formation of the MLH1-PMS2 heterodimer, leading to a negative IHC staining pattern for both proteins. Conversely, the alterations in the *MSH2* gene affect the stability of the MSH2–MSH6 heterodimer with the concurrent IHC loss of MSH2 and MSH6 [3,5,8]. The single loss of MSH6 or PMS2 proteins expression is less commonly observed since the pathogenic variants in the corresponding genes are more rare [9,10,11]. In addition to these molecular mechanisms, the hypermethylation of the *MLH1* gene promoter also appears to play a pivotal role in the development of LS-related cancers. Indeed, a significant proportion of patients with germline pathogenic variants in *MLH1* gene show the epigenetic inactivation of this gene on tumor tissue [12,13]. The occurrence of pathogenic variants within the MMR genes, coupled with hypermethylation of the *MLH1* gene promoter, makes both the interpretation of the IHC expression pattern of MMR proteins and the characterization of the underlying molecular events challenging [14,15]. Here, we present and discuss the case of a patient whose colon rectal cancer (CRC) exhibiting a null IHC staining pattern for four MMR-related proteins offers the opportunity to gain further insights into the specific molecular temporal trajectories contributing to its development.

## 2. Materials and Methods

*MLH1* gene promoter methylation analysis was performed on genomic DNA extracted from both blood and tumor tissue using a SALSA^®^ MS-MLPA^®^ Probemix ME011 Mismatch Repair Genes kit (MRC-Holland, Amsterdam, The Netherlands), which is also able to detect the point mutation c.1799T>A p.(V600E) affecting the *BRAF* gene. In summary, about 100 ng of DNA was hybridized for 16 h at 60 °C with methylation-specific probes containing an HhaI methylation-sensitive digestion site. Next, the probes were processed with an HhaI enzyme, which digests only GCGC unmethylated sequences, and then amplified using PCR with universal FAM-labeled primers. PCR products were run on a SeqStudio genetic analyzer (Thermo Fisher scientific, Waltham, MA, USA) and assessed with GeneMapper software version 5 (Thermo Fisher scientific, Waltham, MA, USA). Data analysis was carried out with Coffalyser.net software v.220513.1739 (MRC-Holland, Amsterdam, The Netherlands). A sample was classified as methylated when CpG sites in the *MLH1* promoter region exhibited methylation higher than a 0.2 ratio, which corresponds to the limit of the blank for each probe. A sample was considered methylated when the mean of all five cytosines was higher than 10% methylation. The 10% cut-off level was determined by analyzing artificial control samples at different percentages of DNA methylation (0%, 10%, 50%, and 100%), which were prepared by mixing commercial fully methylated DNA and fully unmethylated DNA (Human WGA Methylated and Non-methylated DNA Set; Zymo Research). Germline analysis of the MMR genes, including *MLH1, MSH2, MSH6*, and *PMS2*, was performed on blood-derived DNA using standard procedures such as Sanger sequencing and targeted NGS panel sequencing. MLPA testing for the *MSH2* gene was performed using a SALSA^®^ MLPA^®^ Probemix P003-D1 MSH2/MLH1 Kit, which is able to detect large rearrangements in exon 9 of the *EPCAM* gene, and the results were confirmed by a SALSA^®^ MLPA^®^ Probemix P248-B2 MSH2/MLH1 kit. MLPA analysis for the *MSH6* gene was carried out using a SALSA^®^ MLPA^®^ Probemix P072-D1 MSH6-MUTYH Kit, which can also detect large rearrangements in exons 3, 8, 9 e 3’ UTR of the *EPCAM* gene. Data analysis was performed using Coffalyser.net software (MRC-Holland, Amsterdam, The Netherlands). IHC for MMR proteins was carried out with monoclonal antibodies to MLH1 (clone M1, Ventana/Roche, Oro Valley, AZ, USA), MSH2 (clone G219-1129, Cell Marque, Rocklin, CA, USA), MSH6 (clone SP93, Ventana/Roche, Oro Valley, AZ, USA), and PMS2 (clone A16.4, BD Biosciences, Franklin Lakes, NJ, USA).

## 3. Results

The clinical case of the present investigation is in reference to a 65-year-old woman who, at 37 years old, underwent hysteroannesiectomy and extended lymphadenectomy, followed by adjuvant treatment comprising six cycles of cisplatin and cyclophosphamide for bilateral ovarian carcinoma (stage III C) alongside concomitant endometrial cancer (stage IIB). At age 64, during a routine follow-up colonoscopy, an ulcerated vegetating neoformation at the cecum, characterized as intestinal-type adenocarcinoma, was identified. Immunophenotypically, the analysis of the tumor specimen revealed a strong expression of CDX2 and the absence of PAX8 expression, while only sporadic elements exhibited CK20 expression. Subsequently, the patient underwent a right hemicolectomy with a final diagnosis of poorly differentiated adenocarcinoma, pT3LV0N0 0/18 (stage II). The tumor was ulcerated, infiltrating the full thickness of the muscular wall, and initially extending to the perivisceral adipose tissue with an expansive growth pattern and intra- and peritumoral lymphocytic infiltration. Molecular analysis performed on tumor tissue did not detect the c.1799T>A, p.(V600E) variant in the *BRAF* gene, while IHC analysis of MMR proteins revealed an atypical profile characterized by the complete expression loss of all analyzed MMR proteins, indicative of a null IHC staining pattern (Figure 1a–d).

Given the IHC loss of all MMR proteins and the somatic absence of c.1799T>A, p.(V600E) variant in the *BRAF* gene, further in-depth investigations were performed on the tumor tissue with the aim to elucidate the underlying genetic and molecular alterations contributing to the observed phenotypic characteristics. Methylation analysis of tumor tissue conducted on five CpG sites revealed a full hypermethylation of the *MLH1* gene promoter with a percentage ranging from 34% to 41%, while the analysis performed on germline DNA showed the absence of constitutional methylation. Whether or not these results explained the IHC loss of MLH1 and PMS2 protein expression, they did not account for the concurrent loss of MSH2 and MSH6 proteins expression observed in this specific clinical case. Thus, a germline analysis of all four MMR genes was performed, revealing a wide deletion involving the exon 16 of the MSH2 gene. This genetic alteration started from nucleotide 2635: c.(2634+1_2635-1)_(*279+1-?)del p.(?), although its exact extension has not been precisely defined. This deletion is classified as pathogenic variant in the InSiGHT database and is responsible for LS [16].

## 4. Discussion

The epigenetic inactivation of the *MLH1* gene constitutes the cornerstone in the etiology of sporadic CRCs [17]. However, when this alteration is associated with a pathogenic variant in MMR genes, it may become challenging to distinguish between the sporadic and the hereditary nature of cancer [8]. This genetic occurrence should not be underestimated, as 15–17% of patients with CRC exhibiting hypermethylation of the *MLH1* gene promoter also carry a germline variant in the *MLH1* gene [9,10]. Conversely, the epigenetic *MLH1* silencing is much less frequently observed when the pathogenic variant involves the *MSH2*, *MSH6*, and *PMS2* genes [13,18,19,20,21,22,23]. In LS-related CRCs, the concurrent coupled loss of MLH1 and PMS2 or MSH2 and MSH6 expression is generally linked to alterations in the *MLH1* and *MSH2* genes, respectively, while the loss of all four MMR proteins, defined as null IHC staining phenotype, has been only rarely reported [18,24,25]. To the best of our knowledge, only one case of LS-related CRC exhibiting a null IHC staining phenotype has been described [18], which involved a woman who initially developed a right-sided CRC and subsequently a cancer of the ureter. The observed null phenotype was attributed to somatic hypermethylation of the *MLH1* gene promoter associated with a concurrent germline pathogenic variant of the *MSH2* gene consisting of the missense variant c.1759G>C located in exon 11, which resulted in an exon skipping event [18,26,27,28,29]. However, the specific effect of this pathogenic variant on the HIC expression pattern of tumor MMR proteins was not fully elucidated since it may be linked to the absence of MSH2 and MSH6 expression or the loss of expression in all four MMR proteins [18,27]. Conversely, the present study reports a rare case of a woman who was originally diagnosed, at age 37, for ovarian and endometrial cancer, and later at age 64 developed a right-sided CRC exhibiting a null phenotype that was associated with the hypermethylation of the *MLH1* gene promoter, which was thus suggestive of an LS syndrome. Afterwards, in-depth molecular investigations identified a germline large deletion in the *MSH2* gene, resulting in exon 16 loss (Table 1).

The LS-related cancers are generally believed to be driven by a mechanism known as “second hit”, where a somatic pathogenic alteration occurs in the wild-type allele of the affected MMR gene leading to MMR cellular dysfunction triggering the cancer development [30,31,32]. Less frequently, the “second hit” may be attributed to either constitutional or somatic hypermethylation of the *MLH1* gene promoter [12]. The intricate interplay among genetic and epigenetic alterations underscores the multifaceted molecular nature of the development of LS-related cancers and emphasizes the importance of investigating the interrelationships between these two molecular events. In this context, the null phenotype represents an interesting opportunity to gain deeper insight into the temporal events underlying LS-associated tumorigenesis. Two different hypotheses have been previously formulated to explain the molecular mechanism at the base of the null phenotype development in patients who carry *MSH2* constitutional pathogenic variants [13]. The first proposes that somatic biallelic hypermethylation of the *MLH1* gene promoter occurs before the “second hit”, while the second hypothesis suggests that somatic epigenetic inactivation of *MLH1* arises after the *MSH2* somatic mutation. A more in-depth explanation is that when the first event is the biallelic hypermethylation of the *MLH1* gene promoter, it leads to the loss of MLH1 and PMS2 protein expression, while the subsequent loss of MSH2 and MSH6 proteins expression is consequent to *MSH2* gene alteration [13]. Conversely, when biallelic hypermethylation of the *MLH1* gene promoter occurs after the loss of MSH2 and MSH6 proteins expression, it generates the subsequent loss of MLH1 and PMS2 protein expression. In both scenarios, the null IHC phenotype is the ultimate result of two distinct sequences of molecular processes, both linked by the germline pathogenic variant of the *MSH2* gene (Figure 2).

It is crucial to emphasize that in the majority of LS-related cancers diagnosed in patients carrying *MSH2* gene pathogenic variants, the null phenotype represents an exceptionally rare diagnostic occurrence [18]. This particular diagnostic outcome contrasts with the hypothesis that hypermethylation precedes the somatic *MSH2* alteration, as more cases with a null phenotype would be expected. To support this hypothesis, in addition to our case, we reported four LS patients with germline pathogenic variants in the *MSH2* gene, whose CRCs [18,19,22,23] had been investigated for both IHC and epigenetic analyses. Interestingly, the loss of MSH2 protein expression was consistent across all cases, confirming that its loss represents the most sensitive marker for detecting an underlying *MSH2* gene alteration. Instead, the loss of MLH1 expression was observed only in one case, although all cases presented the hypermethylation of the *MLH1* gene promoter (Table 1).

This latter result indicates that in patients carrying *MSH2* pathogenic variants, the somatic hypermethylation of the *MLH1* gene promoter does not always induce a tumor tissue loss of MLH1 protein expression. This aspect is remarkable since in patients who carry *MLH1* gene pathogenic variants, the somatic tumor hypermethylation of the *MLH1* gene promoter is always associated with the loss of MLH1 and PMS2 proteins expression. The apparent incongruence may arise from different molecular mechanisms, including heterogeneity in the cancer methylation pattern. Indeed, not all cancers with hypermethylation of the *MLH1* gene promoter display homogeneous tumor methylation, indicating that this process may not arise from a clonal occurrence [33]. This evidence could be significant, as some CRCs, characterized by the loss of expression of MSH2 and MSH6 proteins, may also exhibit hypermethylation of the *MLH1* gene promoter, but it cannot always be revealed by IHC analysis. Therefore, in carriers of *MSH2* gene pathogenic variants, it is advisable to further characterize the tumor tissue through epigenetic analysis, as the hypermethylation status of the *MLH1* gene promoter may influence patient prognosis [34]. Future studies will be crucial in determining the proportion of *MSH2*-related cancers that also harbor hypermethylation of the *MLH1* gene promoter in order to establish its clinical role in determining the clinical outcomes of CRC patients.

## 5. Conclusions

Taken together, these findings demonstrate that the null IHC phenotype for MMR proteins may indicate the concurrent occurrence of a germline *MSH2* pathogenic variant alongside sporadic hypermethylation of the *MLH1* gene promoter. The latter represents an independent temporal event triggered only at a later stage of cancer development after the occurrence of a double hit, a form of further differentiation in the cancer process. The knowledge of the temporal sequence of the molecular events that lead to the cancer development of *MSH2*-related cancers may have important implications in the clinical management of these CRC patients.

## Figures and Tables

**Figure 1 diagnostics-14-00888-f001:**
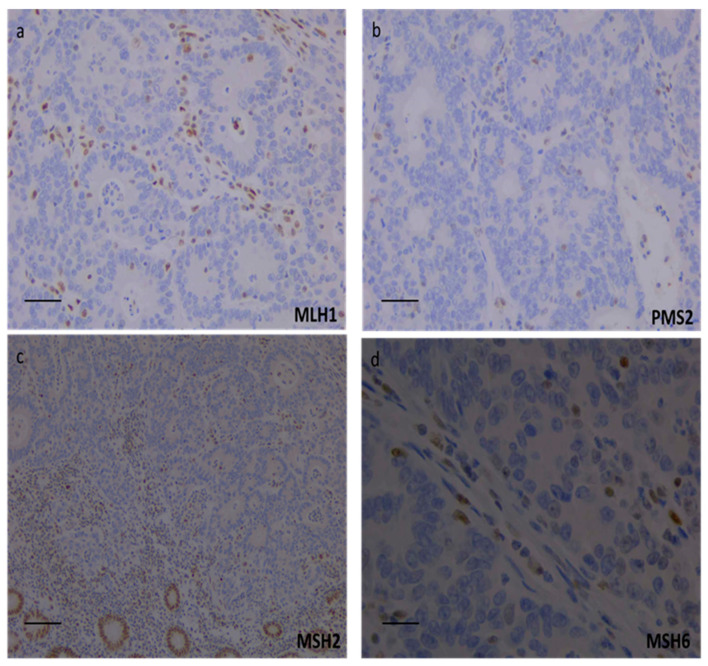
Immunohistochemical evaluation of mismatch repair proteins showed the complete loss of MLH1 (**a**), PMS2 (**b**), MSH2 (**c**), and MSH6 (**d**) proteins expression. The presence of staining in the background lymphocytes and stromal cells represents the positive internal control. (**a**,**b**): scale bar = 400 µm, 20× magnification of camera lens; (**c**): scale bar = 250 µm, 10× magnification; (**d**): scale bar = 650 µm, 40× magnification.

**Figure 2 diagnostics-14-00888-f002:**
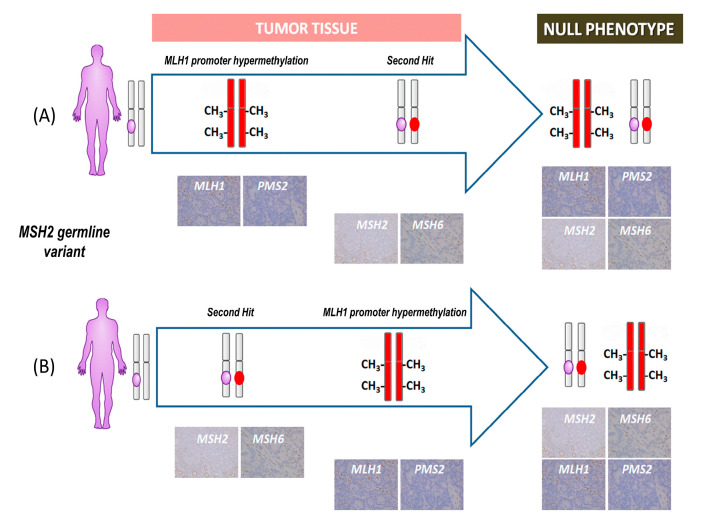
Schematic description of the two hypotheses leading to the formation of the null phenotype. The first hypothesis (**A**) suggests that hypermethylation of the *MLH1* gene promoter occurs before the second hit, while the second hypothesis (**B**) suggests that the second hit occurs before hypermethylation of the *MLH1* gene promoter. *MLH1* gene alleles are depicted in red, while *MSH2* gene alleles are depicted in gray. The purple oval represents the constitutional variant, while the red oval represents the somatic event that characterized the second hit.

**Table 1 diagnostics-14-00888-t001:** CRCs with hypermethylation of the *MLH1* gene promoter derived from patients carrying a pathogenic variant in the *MSH2* gene. The clinical and molecular characteristics are detailed for each case. CRC: colorectal cancer; UC: ureter cancer; OC: ovarian cancer; EC: endometrial cancer; F: female; M: male; NA: not available; R: right; L: left; IHC: immunohistochemical; PM: promoter methylation; (G): germline.

Patient	Cancer Type	Sex	Age at Diagnosis	Side	Location	IHC	MLH1-PM	*BRAF* Mutation	Pathogenic Variant
[18]	CRC	F	63	R	Cecum	MLH1-/MSH2- /MSH6-/PMS2-	C	-	(G)*MSH2*: c.G1759C p.(G587R)
	UC		69						
[19]	CRC	NA	NA	NA	NA	MLH1+/MSH2- /NA/NA	C	NA	(G)*MSH2*: c.A942+3T
[22]	CRC	F	29	L	Descending Colon	MLH1+/MSH2- /NA/NA	C	NA	(G)*MSH2*: c.C1165T p.(R389*)
[22]	CRC	NA	32	NA	NA	MLH1+/MSH2- /NA/NA	B	NA	(G)*MSH2*: c.T524C p.(L175P)
Our Case	CRC OC+EC	F	63 38	R	Cecum	MLH1-/MSH2-/ MSH6-/PMS2-	C	-	(G)*MSH2*: c.(2634+1_2635-1)_(*279+1-?)del p.(?)

## Data Availability

Data are contained within the article.

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
