# Peer review of "Null Mismatch Repair Proteins Expression Reveals the Temporal Molecular Events in Lynch Syndrome-Related Cancers"

_diagnostics, 2024, doi:10.3390/diagnostics14090888_

Round 1
Reviewer 1 Report
Comments and Suggestions for Authors
In their brief report "The null expression phenotype for mismatch repair pro- 2
teins reveals the temporal sequence of genetic and epige- 3
netic alterations in Lynch syndrome-related cancers", Miolo et al describe the rare immunohistochemical detection of all MMR proteins.
The illustrations support the text, but the layout needs to be improved. In particular, the size and shape of the captions need to be adjusted. Table 1 also requires editorial improvement (page break). Apart from this, only improvements in the area of spelling a are necessary. As a brief report, I can recommend the acceptance of the manuscript for publication.
Comments on the Quality of English LanguageMinor improvements should be implemented.
Author Response
Point by point answers of the authors to the reviewers
We really appreciated the time and effort dedicated by reviewers to evaluate our manuscript. Thanks to their comments and suggestions, the results and the overall presentation of the manuscript are significantly improved.
Reviewer 1
The illustrations support the text, but the layout needs to be improved. In particular, the size and shape of the captions need to be adjusted. Table 1 also requires editorial improvement (page break). Apart from this, only improvements in the area of spelling are necessary. As a brief report, I can recommend the acceptance of the manuscript for publication.
Reply: Thank you for your valuable feedback. In the revised version of the manuscript the layout of Figure 1 was improved as suggested by adjusting both the size and the shape of the captions in order to better enhance the visual coherence between the histological images and the text. Additionally, we improved the editorial aspects linked to Table 1, by eliminating the page breaks for better readability.
Comments on the Quality of English Language
Minor improvements should be implemented.
Reply: As suggested, the all manuscript have been revised and corrected for the residual English grammar errors as well as improved for English language and overall reviewed by a native English speaker.
Reviewer 2 Report
Comments and Suggestions for Authors
I have been asked to review the article "The null expression phenotype for mismatch repair proteins reveals the temporal sequence of genetic and epigenetic alterations in Lynch syndrome-related cancers" for Diagnostics, and here are my comments:
1. The authors have labeled the submission as a brief report, but it appears to be a case report. I recommend the authors to clarify this distinction to ensure accurate categorization.
2. The current title is informative but somewhat lengthy and technical. I suggest considering a more concise title that still encapsulates the essence of the research.
3. It's noted that the tense of verbs throughout the article should be past tense to align with the reporting of completed research. For example, "...we present and discuss..." should be "...we presented and discussed..." in the final sentence of the introduction (Line 69).
4. In the results section, if possible, I suggest adding quantification to the immunohistochemical (IHC) results.
5. I recommend enhancing Figure 1 by including additional information regarding the IHC results, possibly through labeling or a legend. Additionally, consider adding a scale bar to the figures for improved visual comprehension.
Comments on the Quality of English LanguageIt's noted that minor revisions for English language clarity and coherence are needed throughout the manuscript.
Author Response
Point by point answers of the authors to the reviewers
We really appreciated the time and effort dedicated by reviewers to evaluate our manuscript. Thanks to their comments and suggestions, the results and the overall presentation of the manuscript are significantly improved.
Reviewer 2
1. The authors have labeled the submission as a brief report, but it appears to be a case report. I recommend the authors to clarify this distinction to ensure accurate categorization.
Reply: We agree with the reviewer that we are describing a single clinical case; however, it is worth to note that IHC null phenotype for MMR proteins associated to a germline pathogenic variant is extremely rare. This latter aspect does not allow to report a large series of cases for meaningful comparison. However, although the study is referred to a single case, it represents an investigation opportunity to compare it with the other rare cases which are available in the literature. On the base of these features we retain that the presented report should be better classified as a brief report.
2. The current title is informative but somewhat lengthy and technical. I suggest considering a more concise title that still encapsulates the essence of the research.
Reply: Taking into account the suggestion of the reviewer we changed the title as followed “The null mismatch repair proteins expression reveals the temporal molecular events in Lynch syndrome-related cancers” which results less technical and more concise maintaining the essence of the research.
3. It's noted that the tense of verbs throughout the article should be past tense to align with the reporting of completed research. For example, "...we present and discuss..." should be "...we presented and discussed..." in the final sentence of the introduction (Line 69).
Reply: Thank for the valuable feedback. The final sentence of the introduction has been changed according to the tenses of the verbs, from present to past as suggested.
4. In the results section, if possible, I suggest adding quantification to the immunohistochemical (IHC) results.
Reply: We have considered the suggestion of reviewer. However, in this specific case it is not possible to add quantification to the immunohistochemical (IHC) results as the expression is completely negative for all IHC markers making any quantification impossible. This aspect was better clarified in result section where, we added the adjective "complete" before the “expression loss".
5. I recommend enhancing Figure 1 by including additional information regarding the IHC results, possibly through labeling or a legend. Additionally, consider adding a scale bar to the figures for improved visual comprehension.
Reply: We have taken into consideration your recommendations and have implemented the Figure 1 by the insertion of the scale bar into the IHC microscopical images for its better visual comprehension. Moreover, in the caption of the figure we specified the dimension parameters expressed by the scale bar (µm) as well as the magnification of the images.
Comments on the Quality of English Language
It's noted that minor revisions for English language clarity and coherence are needed throughout the manuscript.
Reply: As suggested, the manuscript have been revised and corrected for the residual English grammar errors as well as improved for English language and overall reviewed by a native English speaker.